# Nation-wide survey of oral care practice in Japanese intensive care units: A descriptive study

Tomoki Kuribara[1], Takeshi Unoki[1]*, Sachika Yamakita[2], Naoya Hashimoto[3], Yasuyo Yoshino[4], Hideaki Sakuramoto[5], Gen Aikawa[6], Saiko Okamoto[7]

1 Department of Acute and Critical Care Nursing, School of Nursing, Sapporo City University, Sapporo, Hokkaido, Japan, 2 Department of Acute and Critical Care Nursing, School of Nursing, Sapporo City University, Master Program, Sapporo, Hokkaido, Japan, 3 Advanced Critical Care and Emergency Center, Sapporo Medical University Hospital, Sapporo, Hokkaido, Japan, 4 Faculty of Nursing Department of Nursing Adult Nursing, Komazawa Women's University, Inagi-shi, Tokyo, Japan, 5 Department of Critical Care and Disaster Nursing, Japanese Red Cross Kyushu International College of Nursing, Munakata, Fukuoka, Japan, 6 Department of Adult Health Nursing, College of Nursing, Ibaraki Christian University, Omika, Hitachi, Ibaraki, Japan, 7 Department of Nursing, Hitachi General Hospital, Jonancho, Hitachi, Ibaraki, Japan

* iwhyh1029@gmail.com

**Data Availability Statement:** All relevant data are within the paper and its Supporting Information files.

## Abstract

Oral care for critically ill patients helps provide comfort and prevent ventilator-associated pneumonia. However, a standardized protocol for oral care in intensive care units is currently unavailable. Thus, this study aimed to determine the overall oral care practices, including those for intubated patients, in Japanese intensive care units. We also discuss the differences in oral care methods between Japanese ICUs and ICUs in other countries. This study included all Japanese intensive care units meeting the authorities' standard set criteria, with a minimum of 0.5 nurses per patient at all times and admission of adult patients requiring mechanical ventilation. An online survey was used to collect data. Survey responses were obtained from one representative nurse per intensive care unit. Frequency analysis was performed, and the percentage of each response was calculated. A total of 609 hospitals and 717 intensive care units nationwide participated; among these, responses were collected from 247 intensive care units (34.4%). Of these, 215 (87.0%) and 32 (13.0%) reported standardized and non-standardized oral care, respectively. Subsequently, the data from 215 intensive care units that provided standardized oral care were analyzed in detail. The most common frequency of practicing oral care was three times a day (68.8%). Moreover, many intensive care units provided care at unequal intervals (79.5%), mainly in the morning, daytime, and evening. Regarding oral care methods, 96 (44.7%) respondents used only a toothbrush, while 116 (54.0%) used both a toothbrush and a non-brushing method. The findings of our study reveal current oral care practices in ICUs in Japan. In particular, most ICUs provide oral care three times a day at unequal intervals, and almost all use toothbrushes as a common tool for oral care. The results suggest that some oral care practices in Japanese ICUs differ from those in ICUs in other countries.

**Funding:** This work was funded by ALCARE CO., Ltd (grant number: N/A) and TU was received. https://www.alcare.co.jp/en/ The funders had no role in the study design, data collection and analysis, decision to publish, or manuscript preparation.

**Competing interests:** The authors have declared that no competing interests exist.

## Introduction

Providing oral care for critically ill patients is crucial for their comfort, improvement of thirst [1, 2], and preventing ventilator-associated pneumonia (VAP). Oral care helps improve oral health and reduce inflammation [3, 4]. Oral care for intubated patients can reduce the incidence of VAP, ventilator days, and length of intensive care unit (ICU) stay [5]. Nurses in many countries recognize the importance of oral care for such patients [6, 7]. However, a standardized protocol for oral care has yet to be established, with each hospital following its own protocol.

Several countries have conducted studies on the actual situation of oral care in ICUs in the past decade [8, 9]. A study conducted in the United States reported that the main oral care methods were using a foam swab every 2–4 h or a toothbrush every 12 h [8]. Conversely, toothbrushes were used less frequently, and a mouthwash was used every 8–12 h in seven European countries [9]. A Japanese study [10] found that oral care was performed once per shift, with 97% of ICUs using a toothbrush in combination with a foam swab.

Over a decade has passed since the previous study was conducted in Japan, and new oral care guidelines have since been published in other countries [11]. As a result, oral care practices in Japanese ICUs may have changed compared with those in previous studies. In addition, the oral care practices in Japan are likely to be different from those in other countries because the high concentrations of chlorhexidine used in other countries are unavailable in Japan. Literature on the current practice of oral care in Japan, which highlights its differences from those followed in other countries, is lacking. Therefore, this study aimed to determine the overall oral care practices, including those for intubated patients, in ICUs throughout Japan. We also discuss the differences in oral care methods between Japanese ICUs and ICUs in other countries.

## Materials and methods

This cross-sectional descriptive study was conducted using a web-based questionnaire. This study followed the "STROBE" guideline for cross-sectional studies [12] (S1 Text).

### Sample and setting

Japanese ICUs that meet the standards set by the Japanese Ministry of Health, Labour and Welfare (MHLW) are mainly classified into two types: general ICUs and ICUs specializing in emergency medical care. The former includes four types of ICUs with a minimum of 0.5 nurses per patient at all times, while the latter includes ICUs with at least 0.5 nurses per patient at all times and ICUs with at least 0.25 nurses per patient at all times. In Japan, the ICUs with at least 0.5 nurses per patient at all times are major facilities. Thus, in this study inclusion criteria were as follows: (1) ICUs approved by the MHLW with at least 0.5 nurses per patient ratio at all times and (2) ICUs admitting patients requiring mechanical ventilation. Pediatric ICUs were excluded.

Postal mails were sent to nurse directors at all hospitals with eligible ICUs requesting their participation in the survey and referral of one nurse with actual oral care practice in the ICU as the survey respondent. Thus, one response per ICU was received.

### Instrument and data collection

This study used an original questionnaire prepared by all the investigators who are expert researchers having extensive experience in critical care. The questionnaire items were based on previous studies including those conducted in Japan [8, 10, 13]. The final questionnaire items included characteristics of the respondents and their hospitals, assessment of oral

condition, oral care frequency, oral care methods, moisturizing care, preparation of care products, and oral care outcomes (S2 Text). A question on whether oral care was standardized across the unit was included as well. Owing to the nature of our study design, the survey was terminated for ICUs who reported that oral care practices were not standardized. The survey was tested for content validity, comprehensibility, time to complete, and ease of administration by the investigators.

SurveyMonkey (https://www.surveymonkey.com), a security-certified cloud-based tool, was used to create the survey questionnaire. A QR code for the SurveyMonkey questionnaire was attached to the postal mail to the nurse directors. The representative nurse of each eligible ICU accessed the survey via this QR code. The survey period was from January 10, 2023 to February 21, 2023.

### Data analysis

The responses to the questionnaire were analyzed using descriptive analysis. Categorical variables were calculated as the percentage of each response. For continuous variables, the median and interquartile range (IQR) were calculated for non-normal distributions, and the mean and standard deviation (SD) for normal distributions. All statistical analyses were performed by R version 4.2.2 (R Foundation for Statistical Computing, Vienna, Austria).

### Ethical considerations

This study was approved by the Ethics Review Committee of the Sapporo City University (No. 2236–1). The participating nurses were nominated on behalf of their unit; however, they could decide whether to respond to the questionnaire or not. We sent the information of this study with a QR code for the SurveyMonkey questionnaire by postal mail. The first page of the survey form included study information and a section where participants could confirm their consent to take part. To indicate their willingness to participate, individuals submitted their responses and marked a checkbox next to the consent statement. The Ethics Committee reviewed these checkboxes and approved the consent form, considering them as a valid expression of consent.

## Results

The study included 609 hospitals and 717 ICUs nationwide; responses collected from 247 ICUs (34.4%) were included in the analysis. Among these, 32 ICUs (13.0%) reported no standardized oral care practice (non-standardized group) and were excluded from further analysis. Thus, oral care practices in 215 ICUs (87.0%) were analyzed in detail (standardized group).

### Participant characteristics

Characteristics of participating ICUs are listed in Table 1. Regarding participating hospitals, most respondent hospitals had approximately 401–600 beds. In addition, the respondent hospitals had a median of 8 (IQR 6–12) and 7.5 (IQR 6–10) ICU beds for the standardized and non-standardized groups, respectively. S1 Fig illustrates the regional distribution of the 215 ICUs (standardized group) and the number of respondents from each region.

### Assessment of oral condition

Ninety-two (42.8%) ICUs assessed the oral condition immediately after admission; approximately half of them used Eilers' Oral Assessment Guide (OAG). Approximately 46.5% of ICUs indicated that they assessed the oral condition routinely, with 50% of them using the OAG. Details of the responses to oral care condition assessment are listed in Table 2.

**Table 1. Characteristics of participating ICUs.**

| Characteristics | | Overall (n = 247) | Standardized (n = 215) | Non-standardized (n = 32) |
|---|---|---|---|---|
| Total number of hospital beds (%) | | | | |
| | <200 | 8 (3.2) | 6 (2.8) | 2 (6.2) |
| | 200–400 | 67 (27.1) | 58 (27.0) | 9 (28.1) |
| | 401–600 | 79 (32.0) | 68 (31.6) | 11 (34.4) |
| | 601–800 | 53 (21.5) | 48 (22.3) | 5 (15.6) |
| | 801–1000 | 25 (10.1) | 22 (10.2) | 3 (9.4) |
| | 1001–1200 | 13 (5.3) | 11 (5.1) | 2 (6.2) |
| | 1201–1400 | 2 (0.8) | 2 (0.9) | 0 (0.0) |
| Number of ICU beds (median [IQR]) | | 8.0 [6.0, 10.0] | 8.0 [6.0, 12.0] | 7.5 [6.0, 10.0] |
| University hospitals (%) | | 70 (28.3) | 63 (29.3) | 7 (21.9) |
| ICU in emergency department (%) | | 51 (20.6) | 45 (20.9) | 6 (18.8) |
| Staffing of physician in the ICU (%) | | | | |
| | Closed ICU[a] | 13 (5.3) | 11 (5.1) | 2 (6.2) |
| | Semi-closed ICU[b] | 95 (38.5) | 84 (39.1) | 11 (34.4) |
| | Open ICU[c] | 77 (31.2) | 69 (32.1) | 8 (25.0) |
| | No intensivists | 62 (25.1) | 51 (23.7) | 11 (34.4) |
| Number of day-shift weekday-working nurses of (median [IQR]) | | 8.0 [5.0, 10.5] | 8.0 [5.5, 11.0] | 7.0 [5.0, 10.0] |
| Number of night-shift weekday-working nurses (median [IQR]) | | 4.0 [3.0, 6.0] | 4.0 [3.0, 6.0] | 4.0 [3.0, 5.0] |

Abbreviations: ICU, intensive care unit; IQR, interquartile range

[a]Intensivist is the attending physician.

[b]Intensivists are not attending physicians; however, all patients admitted to the ICU are visited by an intensivist.

[c]Intensivists are involved in patient care only when the attending physician requests to see them.

**Table 2. Response to oral condition assessment.**

| Survey item | | Value (n = 215 ICUs) |
|---|---|---|
| Oral evaluation is performed using a scale immediately upon ICU admission (%) | | 92 (42.8) |
| Type of scale used (%)[a] | | |
| | OAG | 42 (45.7) |
| | ROAG | 12 (13.0) |
| | OHAT-J | 27 (29.3) |
| | Others | 11 (12.0) |
| Oral evaluations are routinely performed using a scale (%) | | 100 (46.5) |
| Type of scale used (%)[b] | | |
| | OAG | 50 (50.0) |
| | ROAG | 12 (12.0) |
| | OHAT-J | 28 (28.0) |
| | Others | 10 (10.0) |

Abbreviations: ICU, intensive care unit; OAG, oral assessment guide; ROAG, revised oral assessment guide; OHAT-J, oral health assessment tool–Japan

[a]Only those who responded that they evaluate the oral status using a scale immediately upon ICU admission (n = 92).

[b]Only those who responded that they routinely evaluate the oral status using a scale (n = 100).

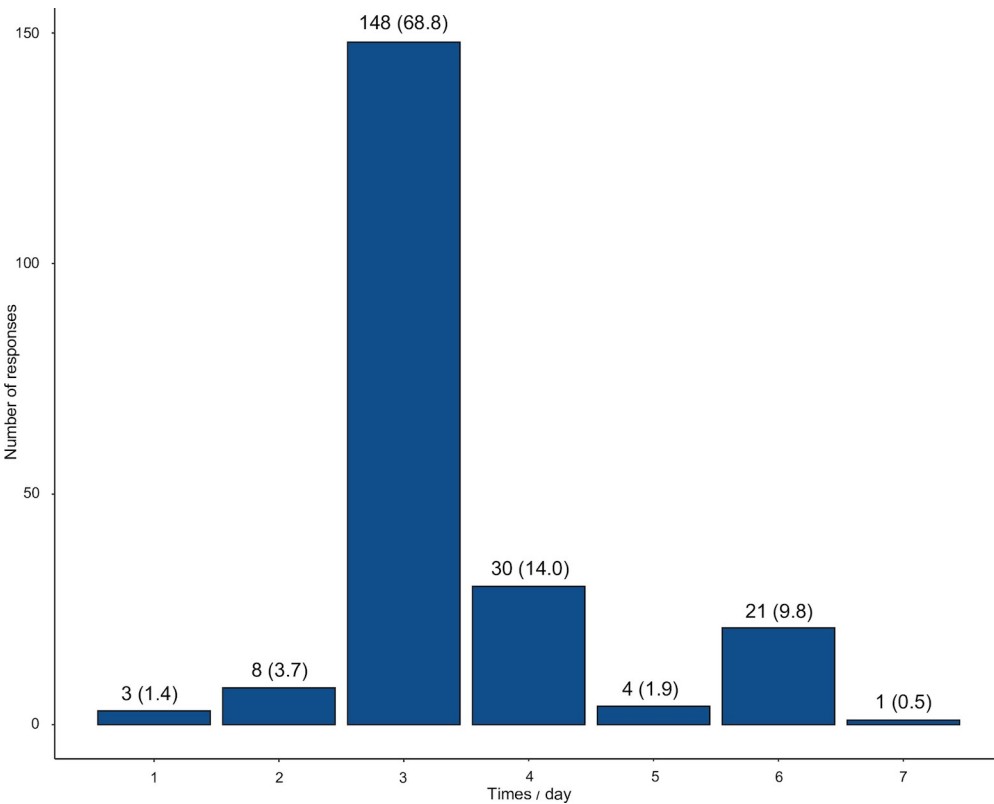

**Fig 1. Oral care frequency.** The number of responses indicating the frequency of oral care sessions per day is represented by n (%).

## Oral care frequency

Oral care was typically provided three times a day (68.8%) (Fig 1). Many ICUs implemented unequal intervals for oral care (79.5%), mainly in the morning, daytime, and evening. Conversely, ICUs providing care at equal intervals typically provided oral care every 4 or 6 h. Among ICUs that provided care at unequal intervals, some provided care in the morning, daytime, and evening, while others provided care in the morning, daytime, before sleep, or at all of these times. The most common reasons for administering care at unequal intervals were the need to adapt to daily life, work systems, and limited human resources. Details of the responses to oral care frequency are listed in S1 Table.

## Oral care methods

Approximately 87.9% of respondents reported adjusting cuff pressure, with a median pressure adjustment of 30 mmHg (IQR 25–30) before providing oral care to intubated patients. When providing oral care, 96 (44.7%) respondents used only a toothbrush, 116 (54.0%) used both a toothbrush and a non-brushing method, and three (1.4%) used only a non-brushing method. Adult toothbrushes were mostly used. Approximately half of the respondents rinsed the patient's mouth after brushing, most commonly with water (median volume, 20 mL). Details of the 212 respondents who used toothbrushes for oral care and their responses regarding other oral care methods are listed in Table 3. In addition, details of responses to non-brushing oral care are listed in S2 Table.

**Table 3. Response to toothbrushing and other methods of oral care.**

| Survey item | | Value (n = 212 ICUs) |
|---|---|---|
| Adjusting cuff pressure before oral care (%)[a] | | 189 (87.9) |
| Pressure for adjusting cuff (median [IQR]) [b] | | 30.0 [25.0, 30.0] |
| Frequency of toothbrushing, times/day (%)[c] | | |
| | Once | 28 (24.1) |
| | Twice | 29 (25.0) |
| | Three times | 57 (49.1) |
| | Four times | 2 (1.7) |
| Toothbrush type (%) | | |
| | Adult | 208 (98.1) |
| | Child | 3 (1.4) |
| | Electric toothbrush | 1 (0.5) |
| Single-use disposable toothbrush (%) | | 51 (24.1) |
| Using toothpaste (%) | | 60 (28.3) |
| Rinsing after brushing (%) | | 112 (52.8) |
| Amount of liquid used for rinsing after brushing, mL/time (%)[d] | | 20.0 [10.0, 50.0] |
| Liquid mainly used after brushing (%)[d] | | |
| | Water | 84 (75.0) |
| | Sterile water | 5 (4.5) |
| | Mouthwash (over the counter or included in kit) | 12 (10.7) |
| | Povidone-Iodine | 1 (0.9) |
| | Others | 10 (8.9) |

Abbreviations: IQR, interquartile range

[a]Response from all participants (n = 215).

[b]Only those who responded that they adjust the cuff pressure before oral care (n = 189).

[c]Only those who responded that they combine toothbrush and non-toothbrush methods for oral care (n = 116).

[d]Only those who responded that rinsing after brushing is performed (n = 112).

## Oral moisturizers

Approximately 55.3% of respondents reported routinely using some form of oral moisturizers. The products used varied among the respondents; 47.0% reported routinely using lip moisturizers (most of which were prescribed by the hospital), while 14.9% reported using antiseptic gels (S3 Table).

## Discussion

The findings of this study revealed current oral care practices in ICUs in Japan. In particular, most ICUs provided oral care three times a day at unequal intervals, mainly in the morning, daytime, and evening, and almost all used toothbrushes as a common tool for oral care, with water being the predominant choice for rinsing the mouth after brushing. Approximately half of ICUs used oral assessment tools or provided moisturizing care in addition to toothbrushes.

The present study revealed that, in Japan, less than half of ICUs routinely used oral assessment tools during ICU admission. Many Japanese ICUs have a higher usage of oral assessment tools than do ICUs in other countries. A study revealed that 35% of nurses in the United Kingdom used oral assessment tools during ICU admission [14], and another study reported that most nurses in the United States did not use oral assessment tools [8]. The current state of oral

care in Japanese ICUs, as found in our study, shows the difference in the rate of use of oral assessment tools compared to that reported in ICUs in the United Kingdom and the United States. However, there may be insufficient repeated use of assessment tools compared to international consensus, the same as the United Kingdom and the United States. The British Association of Critical Care Nurses (BACCN) provided a consensus statement for oral care in 2021 and recommended that oral condition should be assessed with a standardized tool within 6 h of ICU admission and repeated within 12 h at least [11]. The oral assessment tools not only assess the oral condition but also the comfort of the patient. Thus, using the assessment tool to ensure validity may improve the quality of life for patients during hospitalization.

According to our results, oral care in Japanese ICUs was not performed at fixed intervals, such as 2–4 h; rather, it depended on the work shift. The BACCN has recommended oral care be provided every 2–4 h [11], and many previous studies in other countries reported that oral care was provided every 2–4 h [7, 8, 13, 15]. In this study, the most common response regarding oral care frequency was that it was provided three times a day at unequal intervals because of issues such as adapting to the patient's lifestyle, which may be unique to Japanese culture, and human resource limitations. Therefore, oral care was provided less frequently in Japanese ICUs than in ICUs in other countries, indicating the need to re-evaluate our perception of oral care frequency.

Brushing is more frequently implemented in Japanese ICUs than in ICUs in other countries. Brushing twice daily is recommended for plaque removal [11, 16]. Moreover, oral cleansing with swabs and other tools is recommended for every 2–4 h [11]. Previous studies in other countries have reported that both brushing and foam swabs are used [8, 13, 15, 17, 18]. A recent study reported that brushing was most commonly used for oral care (41%) [14]. In this study, almost all Japanese ICUs used brushing for oral care. Although brushing and swabbing are not performed routinely, brushing three times daily is performed in many Japanese ICUs. The results suggest that oral care methods in Japanese ICUs differ from those in ICUs in other countries.

In Japanese ICUs, the use of chlorhexidine on the mucosa at concentrations similar to those used overseas is prohibited, and water is mainly used as a rinsing solution after brushing. The BACCN recommends rinsing after brushing using a small syringe (e.g., 5 mL), as even a small amount of oral rinse can remove contamination [11, 19], and routine use of chlorhexidine is not recommended [11]. Recommendations for the optimal amount of rinsing solution are expected in the future.

The present study indicated that moisturizing care was performed less frequently in Japanese ICUs than in ICUs in other countries, with approximately half of the respondents providing moisturizing care. In other countries, approximately 90% of nurses provided moisturizing care [7]. Moisturizing the mouth and lips after oral care is recommended [11]. The results suggest that provision of such care may need to be practiced more actively.

This study has some limitations. First, selection bias was a possibility. However, responses were obtained from a wide geographic area of Japan, which we believe may not have significantly affected our findings. Second, the responses were subjective. The answers comprised subjective options; therefore, the accuracy of the answers depended on each participant.

## Research implication

The questionnaire survey may not provide a precise reflection of actual oral care practices in Japanese ICUs. Thus, a one-point prevalence study could be instrumental in elucidating the genuine state of oral care. Moreover, the findings of this study suggest disparities between oral care methods in Japanese ICUs and ICUs in other countries. These results also underscore the

influence of traditional cultures and lifestyles on varying oral care practices in different regions and countries. Consequently, the absence of a universally accepted standard for oral care is evident, and there is a need for a clearer understanding of the impact of various oral care approaches in preventing VAP. This suggests the necessity for a randomized controlled trial to establish concrete evidence regarding the efficacy of different oral care methods.

## Conclusions

In summary, most ICUs in Japan provide oral care three times a day at unequal intervals, and almost all use toothbrushes as a common tool for oral care. The results suggest that some oral care practices in Japanese ICUs differ from those in ICUs in other countries. Future studies should focus on the efficacy of different oral care methods to provide more evidence.

## Supporting information

**S1 Fig. Distribution of participants in Japan.**
(DOCX)

**S1 Table. Responses to oral care frequency.**
(DOCX)

**S2 Table. Responses to non-brushing oral care.**
(DOCX)

**S3 Table. Responses to moisturizing care.**
(DOCX)

**S1 Text. STROBE statement.**
(DOCX)

**S2 Text. Questionnaire.**
(DOCX)

## Acknowledgments

We would like to thank everyone who participated in this study. We would like to thank Editage (www.editage.com) for English language editing.

## Author Contributions

**Conceptualization:** Tomoki Kuribara, Takeshi Unoki, Sachika Yamakita, Naoya Hashimoto, Yasuyo Yoshino, Hideaki Sakuramoto, Gen Aikawa, Saiko Okamoto.

**Data curation:** Tomoki Kuribara, Takeshi Unoki, Sachika Yamakita, Naoya Hashimoto.

**Formal analysis:** Tomoki Kuribara, Takeshi Unoki.

**Funding acquisition:** Takeshi Unoki.

**Investigation:** Tomoki Kuribara, Takeshi Unoki.

**Methodology:** Tomoki Kuribara, Takeshi Unoki, Sachika Yamakita, Naoya Hashimoto, Yasuyo Yoshino, Hideaki Sakuramoto, Gen Aikawa, Saiko Okamoto.

**Project administration:** Takeshi Unoki.

**Supervision:** Takeshi Unoki.

**Validation:** Tomoki Kuribara, Takeshi Unoki.

**Visualization:** Tomoki Kuribara, Takeshi Unoki.

**Writing – original draft:** Tomoki Kuribara, Takeshi Unoki.

**Writing – review & editing:** Tomoki Kuribara, Takeshi Unoki, Sachika Yamakita, Naoya Hashimoto, Yasuyo Yoshino, Hideaki Sakuramoto, Gen Aikawa, Saiko Okamoto.

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
