## [Decision Letter · Decision Letter 0]

2 Oct 2023

PONE-D-23-23882Nation-Wide Survey of Oral Care Practice in Japanese Intensive Care Unit : a Descriptive StudyPLOS ONE

Dear Dr. Unoki,

Thank you for submitting your manuscript to PLOS ONE. After careful consideration, we feel that it has merit but does not fully meet PLOS ONE’s publication criteria as it currently stands. Therefore, we invite you to submit a revised version of the manuscript that addresses the points raised during the review process.

We look forward to receiving your revised manuscript.

Kind regards,

Yujiro Matsuishi

Academic Editor

PLOS ONE

Journal Requirements:

3. Thank you for stating the following in the Competing Interests: 

   "I have read the journal's policy and the authors of this manuscript have the following competing interests: Dr. Unoki has received research grants  from Alcare Co. Ltd.  The remaining authors report no conflicts of interest. "

We note that one or more of the authors have an affiliation to the commercial funders of this research study : Alcare Co. Ltd. 

Additional Editor Comments:

Dear Dr. Unoki,    

We are interested in publishing your work, but we believe it requires more fully addressing comments before reconsideration for publication in the journal. My comments are included below for your attention.

＿＿＿＿＿＿＿＿＿＿＿＿＿＿＿＿＿

One reviewer noted some concerns regarding the generalisability of the study. This point should be addressed in more depth in the discussion.

In particular, the importance of the study's publication in an international scientific journal should be mentioned in the discussion. The current situation in Japan is discussed in depth and is of great significance here. However, please add what implications the results of the Japanese study have for practice in other countries, such as the USA and the European bloc.

These opinions and changes will be passed on to a different Reviewwe than the Reviewer who made the decision to Reject.

Please add an additional note regarding the international generalisability of this research.

＿＿＿＿＿＿＿＿＿＿＿＿＿＿＿

Kind regards,    

Yujiro Matsuishi

Academic Editor

Reviewers' comments:

Reviewer's Responses to Questions

**Comments to the Author**

1. Is the manuscript technically sound, and do the data support the conclusions?

Reviewer #1: Yes

Reviewer #2: Partly

2. Has the statistical analysis been performed appropriately and rigorously? 

Reviewer #1: N/A

Reviewer #2: Yes

3. Have the authors made all data underlying the findings in their manuscript fully available?

Reviewer #1: Yes

Reviewer #2: Yes

4. Is the manuscript presented in an intelligible fashion and written in standard English?

Reviewer #1: Yes

Reviewer #2: Yes

5. Review Comments to the Author

Reviewer #1: Title: Nation-Wide Survey of Oral Care Practice in Japanese Intensive Care Unit : a Descriptive Study

Manuscript number: PONE-D-23-23882

This was a cross-sectional (survey) study evaluating the practices of oral hygeine in criticaly ill patients in Japan.

I would like to thank and commend the authors for a job very well done.

The artice is well written, the topic is of great importance, and the manuscript incudes all the structural components of an article.

I have no further comments, and recommend acceptance.

Reviewer #2: Thank you for the invitation to review this manuscript.

The author present the results of an online survey of oral nursing care in Japanese ICUs. Based upon the limited response they received, the authors believe that practicies in Japan differ from those reported in the US and Europe.

1. This survey does not meet the first PLOS ONE criteria for publication, namely, "the results of primary scientific research.

2. There is an absense of evidence that oral care in orally / nasally intubated patients has any patient-centred measureable outcome benefit. As a comfort measure however, it has obvious value.

3. What does it change knowing that that there are somewhat different ICU nursing practices in different coutries in regard to oral care in intubated patients?

6. PLOS authors have the option to publish the peer review history of their article (what does this mean?). If published, this will include your full peer review and any attached files.

Reviewer #1: No

Reviewer #2: **Yes: **Jonathan Ball

---

## [Author Response · Author response to Decision Letter 0]

28 Nov 2023

Response to Editor and Reviewers

Journal Requirements

Comment 1

Response

Thank you for the comments. We have revised the manuscript and ensured that it conforms with the PLOS ONE style. 

Comment 2

Please provide additional details regarding participant consent. In the ethics statement in the Methods and online submission information, please ensure that you have specified what type you obtained (for instance, written or verbal, and if verbal, how it was documented and witnessed). If your study included minors, state whether you obtained consent from parents or guardians. If the need for consent was waived by the ethics committee, please include this information.

Response

Thank you for pointing this out. In this study, the Ethics Committee had approved a consent form waiver. We have described the details of which as follows:

Page 5, Line 104 - L109

We sent the information of this study with a QR code for the SurveyMonkey questionnaire by postal mail. The first page of the survey form included study information and a section where participants could confirm their consent to take part. To indicate their willingness to participate, individuals submitted their responses and marked a checkbox next to the consent statement. The Ethics Committee reviewed these checkboxes and approved the consent form, considering them as a valid expression of consent.

Comment 3

Thank you for stating the following in the Competing Interests: 

 "I have read the journal's policy and the authors of this manuscript have the following competing interests: Dr. Unoki has received research grants from Alcare Co. Ltd. The remaining authors report no conflicts of interest. "

We note that one or more of the authors have an affiliation to the commercial funders of this research study : Alcare Co. Ltd. 

Within your Competing Interests Statement, please confirm that this commercial affiliation does not alter your adherence to all PLOS ONE policies on sharing data and materials by including the following stateament: "This does not alter our adherence to PLOS ONE policies on sharing data and materials.” (as detailed online in our guide for authors http://journals.plos.org/plosone/s/competing-interests). If this adherence statement is not accurate and there are restrictions on sharing of data and/or materials, please state these. Please note that we cannot proceed with consideration of your article until this information has been declared.

Response

Thank you for your comment. We received our research funding from Alcare Co., but the company did not pay salaries to any authors and is not fully involved in the research. Therefore, we sent an enquiry e-mail to Nicholas Wrigley, Publications Assistant and received a confirmation regarding this information. 

The company did not play a role in study design, data collection and analysis, publication decisions, or manuscript preparation. None of the remaining authors have an affiliation with the funders of this research study. 

Additional Editor Comments

One reviewer noted some concerns regarding the generalisability of the study. This point should be addressed in more depth in the discussion.

In particular, the importance of the study's publication in an international scientific journal should be mentioned in the discussion. The current situation in Japan is discussed in depth and is of great significance here. However, please add what implications the results of the Japanese study have for practice in other countries, such as the USA and the European bloc.

These opinions and changes will be passed on to a different Reviewwe than the Reviewer who made the decision to Reject.

Please add an additional note regarding the international generalisability of this research.

Response

Thank you for your comments. We revised our manuscript following your comments. We additionally describe the importance of this study for international science and generalizability in the discussion section.

Page 13 - 14, Line 230 - L 237

The questionnaire survey may not provide a precise reflection of actual oral care practices. Thus, a one-point prevalence study could be instrumental in elucidating the genuine state of oral care. Moreover, the findings from this study revealed disparities between Japanese oral care methods and those from other countries. These results also underscore the influence of traditional cultures and lifestyles on varying oral care practices in different regions and countries. Consequently, the absence of a universally accepted standard for oral care is evident, and there is a need for a clearer understanding of the impact of various oral care approaches in preventing VAP. This suggests the necessity for a randomized controlled trial to establish concrete evidence regarding the efficacy of different oral care methods.

Reviewers’ comments

Reviewer #1

Comment

This was a cross-sectional (survey) study evaluating the practices of oral hygeine in criticaly ill patients in Japan.

I would like to thank and commend the authors for a job very well done.

The artice is well written, the topic is of great importance, and the manuscript incudes all the structural components of an article.

I have no further comments, and recommend acceptance.

Response

We greatly appreciate the insightful comments on our manuscript.

Reviewer #2

Thank you for the invitation to review this manuscript.

The author present the results of an online survey of oral nursing care in Japanese ICUs. Based upon the limited response they received, the authors believe that practicies in Japan differ from those reported in the US and Europe.

Comment 1

This survey does not meet the first PLOS ONE criteria for publication, namely, "the results of primary scientific research.

Response

Thank you for your valuable review and comments. We have carefully reviewed the PLOS ONE publication criteria and believe our study constitutes primary scientific research. In support of this assertion, we followed a rigorous academic methodology in line with the STROBE statement, conducting a nationwide study in Japan that encompassed hospitals all over Japan.

 Although guidelines for oral care have been published, this is one example of how oral care varies from culture to culture. We believe this study is valuable for creating a global standard of oral care.

Comment 2

There is an absense of evidence that oral care in orally / nasally intubated patients has any patient-centred measureable outcome benefit. As a comfort measure however, it has obvious value.

Response

Thank you for your insightful comment. As you indicated, we have revised the discussion in the manuscript as follows, considering the importance of assessing the patient comfort with the assessment tool.

Page 12, Line 193 - L 198

However, our findings may be insufficient according to international consensus. The British Association of Critical Care Nurses (BACCN) provided a consensus statement for oral care in 2021 and recommended that oral condition should be assessed with a standardized tool within 6 h of ICU admission and repeated within 12 h at least [9]. The oral assessment tools not only assess the oral condition but also the comfort of the patient. Thus, using the assessment tool to ensure validity may improve the quality of life for patients during hospitalization.

Comment 3

What does it change knowing that that there are somewhat different ICU nursing practices in different coutries in regard to oral care in intubated patients?

Response

Thank you for your important comment. We should have described the generalizability of the study sufficiently. We have revised the research implication in the manuscript follows:

Page 13 - 14, Line 230 - L 237

The questionnaire survey may not provide a precise reflection of actual oral care practices. Thus, a one-point prevalence study could be instrumental in elucidating the genuine state of oral care. Moreover, the findings from this study revealed disparities between Japanese oral care methods and those from other countries. These results also underscore the influence of traditional cultures and lifestyles on varying oral care practices in different regions and countries. Consequently, the absence of a universally accepted standard for oral care is evident, and there is a need for a clearer understanding of the impact of various oral care approaches in preventing VAP. This suggests the necessity for a randomized controlled trial to establish concrete evidence regarding the efficacy of different oral care methods.

---

## [Decision Letter · Decision Letter 1]

11 Jan 2024

PONE-D-23-23882R1Nation-Wide Survey of Oral Care Practice in Japanese Intensive Care Unit : a Descriptive StudyPLOS ONE

Dear Dr. Unoki,

Thank you for submitting your manuscript to PLOS ONE. After careful consideration, we feel that it has merit but does not fully meet PLOS ONE’s publication criteria as it currently stands. Therefore, we invite you to submit a revised version of the manuscript that addresses the points raised during the review process.

Yujiro Matsuishi

Academic Editor

PLOS ONE

Additional Editor Comments:

Dear Author,

Thank you very much for submitting this revised paper.

One Reveiwer has been deemed Reject and one Major Revision.

As one reviewer has already judged the paper as Accept, we will judge it as Major Revision.

・Please correct the points raised by the reviewer who judged it as Major Revision. please refer to the opinions of the reviewer who judged it as Reject if they can be queried.

・Please make every effort to refer to the opinions of Reviewers who have been judged as Reject if you can.

・Please note that the Reviewer has stated that he/she will leave it to the Editor to decide on the significance of submitting to PLOS ONE for research conducted in Japan.

　We believe that there is no problem with the Editor's decision on this point. Please reply to this matter by stating that the "Editor decided that there is no problem".

Reviewers' comments:

Reviewer's Responses to Questions

**Comments to the Author**

1. If the authors have adequately addressed your comments raised in a previous round of review and you feel that this manuscript is now acceptable for publication, you may indicate that here to bypass the “Comments to the Author” section, enter your conflict of interest statement in the “Confidential to Editor” section, and submit your "Accept" recommendation.

Reviewer #3: (No Response)

Reviewer #4: (No Response)

2. Is the manuscript technically sound, and do the data support the conclusions?

Reviewer #3: No

Reviewer #4: No

3. Has the statistical analysis been performed appropriately and rigorously? 

Reviewer #3: N/A

Reviewer #4: Yes

4. Have the authors made all data underlying the findings in their manuscript fully available?

Reviewer #3: Yes

Reviewer #4: Yes

5. Is the manuscript presented in an intelligible fashion and written in standard English?

Reviewer #3: No

Reviewer #4: Yes

6. Review Comments to the Author

Reviewer #3: General comments

The authors reported the frequency and methods of oral care in patients admitted to the intensive care units (ICU) in Japan. They found that the most common frequency of oral care was thrice a day (68.8%), and most ICUs provided the care at unequal intervals in a day.

They conclude that oral care practices in Japan differ from those in other countries, especially in frequency and methods.

This article's topic may be valuable for the field of intensive care in terms of preventing ventilator-associated pneumonia.

However, from the reviewer's perspective, this article is just a report, not original research. In conclusion, the authors claimed that oral care practices in Japan differs from those in other countries without direct data collection from those countries. Their original data, which is only collected from Japanese ICUs, does not support their conclusion.

These data are valuable to publicize elsewhere, but this is not suitable for publicizing as an original research article, at least in the current form.

Minor comments

P4L74: Is it true the authors included all ICUs in Japan? Please describe the definition of ICU, and what list the authors used to pick up ICUs in hospitals in Japan.

Table 1: Are the characteristics of respondents (e.g., gender, designation, nursing experience, etc.) necessary? Do these characteristics affect something in the results?

Reviewer #4: Thank you for allowing me to review "Nation-Wide Survey of Oral Care Practice in Japanese Intensive Care Unit : a Descriptive Study" by Kuribara, Unoki et al.

The authors performed survey of oral care for intubated patients in Japanese ICU.

This is an interesting study. However, it is a difficult question whether a study that merely reports the results of this Japan-only survey is appropriate for an international journal. I leave it to the editorial board to decide, but a letter might be more appropriate.

My suggestions are as follows.

Abstract

“A total of 609 hospitals and 717 ICUs nationwide participated; among these, 215 (30.0%) and 32 (13%) reported standardized and non-standardized oral care, respectively”

215/717=30%

However, 32/717=4.4%

Please confirm.

I think that responses were collected from 247 ICUs.

This information should be included in the abstract.

“Regarding oral care methods, 96 (38.9%) respondents used only a toothbrush, while 116 (46.9%) used both a toothbrush and a non-brushing method.”

How much is the denominator?

It is confusing.

“differed from those in other countries.”

How different?

Introduction

“Providing oral care for critically ill patients is crucial for their comfort”

Please provide references.

“However, a standardized protocol for oral care is yet to be established, with each hospital following its own protocol.”

This sentence should be moved to the previous paragraph.

Materials and Methods

Data analysis

“The responses to the questionnaire were analyzed by calculating the frequency and percentage of each 97 response.”

What is the difference between the frequency and percentage?

Results

Participant characteristics

I think information such as gender, experience and designation of the responding nurses is unnecessary for the results and Table1.

Table 1. Participant characteristics

“Intensivist is the attending physician”, “Intensivists are not attending physicians; however, all patients admitted to the ICU are seen by an intensivist”, “Intensivists are involved in patient care only when the attending physician requests to see them”

The explanation is a bit redundant; why not state closed ICU (J Intensive Care. 2018 Sep 3:6:57.), semi closed ICU, open ICU, etc., and put the explanation below the table?

Table 2. Response to oral condition assessment

“Frequency”

The number of the hospital (%)?

Table 3. Response to toothbrushing and other methods of oral care

Same as above.

Discussion

“Furthermore, the differences and similarities between Japanese ICUs and those in other countries were highlighted.”

The differences between Japanese ICUs and ICUs in other countries was described, but what are the similarities?

“However, our findings may be insufficient according to international consensus.”

It is difficult to understand the meaning of this sentence.

Do authors mean “However, the current state of oral care in Japan, as found in our study, may be insufficient according to the international consensus.”

7. PLOS authors have the option to publish the peer review history of their article (what does this mean?). If published, this will include your full peer review and any attached files.

Reviewer #3: No

Reviewer #4: **Yes: **Jun Takeshita

---

## [Author Response · Author response to Decision Letter 1]

2 Feb 2024

Response to Reviewers

Dear Editor and Reviewers

Thank you very much for reviewing our manuscript and offering valuable advice. We have addressed your comments with point-by-point responses and revised the manuscript accordingly.

Comments from Academic Editor

・Please correct the points raised by the reviewer who judged it as Major Revision. please refer to the opinions of the reviewer who judged it as Reject if they can be queried.

・Please make every effort to refer to the opinions of Reviewers who have been judged as Reject if you can.

・Please note that the Reviewer has stated that he/she will leave it to the Editor to decide on the significance of submitting to PLOS ONE for research conducted in Japan.

　We believe that there is no problem with the Editor's decision on this point. Please reply to this matter by stating that the "Editor decided that there is no problem".

Response

We appreciate your helpful comment on our manuscript. We have revised our manuscript, addressing the major revision points raised by the reviewers and incorporating suggestions from those who recommended rejection to the best of our ability.

Comments from Reviewer #3: 

General comments

The authors reported the frequency and methods of oral care in patients admitted to the intensive care units (ICU) in Japan. They found that the most common frequency of oral care was thrice a day (68.8%), and most ICUs provided the care at unequal intervals in a day.

They conclude that oral care practices in Japan differ from those in other countries, especially in frequency and methods.

This article's topic may be valuable for the field of intensive care in terms of preventing ventilator-associated pneumonia.

However, from the reviewer's perspective, this article is just a report, not original research. In conclusion, the authors claimed that oral care practices in Japan differs from those in other countries without direct data collection from those countries. Their original data, which is only collected from Japanese ICUs, does not support their conclusion.

These data are valuable to publicize elsewhere, but this is not suitable for publicizing as an original research article, at least in the current form.

Response

We appreciate your helpful comments on our manuscript. We have revised our manuscript based on your suggestions. In addition, the editor has informed us about the general comments of reviewer #3, stating that “Editor decided that there is no problem.”

Minor comments

Comment

P4L74: Is it true the authors included all ICUs in Japan? Please describe the definition of ICU, and what list the authors used to pick up ICUs in hospitals in Japan.

Response

Thank you for your insightful review of our paper. We regret that our manuscript contained inaccuracies and lacked a detailed description of the sample and setting section. Therefore, we have added a detailed description of the sample and setting section and revised the abstract as follows:

Page 2, Line 26 – L 30

Thus, this study aimed to determine the overall oral care practices, including those for intubated patients, in Japanese intensive care units. This study included all Japanese intensive care units meeting the authorities' standard set criteria, with a minimum of 0.5 nurses per patient at all times and admission of adult patients requiring mechanical ventilation. An online survey was used to collect data.

Page 4, Line 77 – L 87

Japanese ICUs that meet the standards set by the Japanese Ministry of Health, Labour and Welfare (MHLW) are mainly classified into two types: general ICUs and ICUs specializing in emergency medical care. The former includes four types of ICUs with a minimum of 0.5 nurses per patient at all times, while the latter includes ICUs with at least 0.5 nurses per patient at all times and ICUs with at least 0.25 nurses per patient at all times. In Japan, the ICUs with at least 0.5 nurses per patient at all times are major facilities. Thus, in this study inclusion criteria were as follows: (1) ICUs approved by the MHLW with at least 0.5 nurses per patient ratio at all times and (2) ICUs admitting patients requiring mechanical ventilation. Pediatric ICUs were excluded.

Postal mails were sent to nurse directors at all hospitals with eligible ICUs requesting their participation in the survey and referral of one nurse with actual oral care practice in the ICU as the survey respondent. Thus, one response per ICU was received.

Comment

Table 1: Are the characteristics of respondents (e.g., gender, designation, nursing experience, etc.) necessary? Do these characteristics affect something in the results?

Response

Thank you for pointing this out. We have removed gender, designation, nursing experience, nursing experience at current hospital, and nursing experience in ICU from the Results section and Table 1 based on your suggestion. In addition, we have revised the title of Table 1, please see Table 1.

Page 6, Line 128 – L 131

Characteristics of participating ICUs are listed in Table 1. Regarding participating hospitals, most respondent hospitals had approximately 401–600 beds. In addition, the respondent hospitals had a median of 8 (IQR 6–12) and 7.5 (IQR 6–10) ICU beds for the standardized and non-standardized groups, respectively.

Page 6, Line 134

Table 1. Characteristics of participating ICUs

Comments from Reviewer #4: 

Thank you for allowing me to review "Nation-Wide Survey of Oral Care Practice in Japanese Intensive Care Unit : a Descriptive Study" by Kuribara, Unoki et al.

The authors performed survey of oral care for intubated patients in Japanese ICU.

This is an interesting study. However, it is a difficult question whether a study that merely reports the results of this Japan-only survey is appropriate for an international journal. I leave it to the editorial board to decide, but a letter might be more appropriate.

My suggestions are as follows.

Response

We appreciate your helpful comments. Please see our responses to each suggestion below:

Abstract

Comment

“A total of 609 hospitals and 717 ICUs nationwide participated; among these, 215 (30.0%) and 32 (13%) reported standardized and non-standardized oral care, respectively”

215/717=30%

However, 32/717=4.4%

Please confirm.

I think that responses were collected from 247 ICUs.

This information should be included in the abstract.

Response

Thank you for pointing this out. We are very sorry. The denominator was incorrect for some of the values during calculation. In this sentence, we set “247” as the denominator. Therefore, we have reviewed and revised the values in the abstract and main text to accurately reflect the correct values.

Page 2, Line 32 – L 36

A total of 609 hospitals and 717 intensive care units nationwide participated; among these, responses were collected from 247 intensive care units (34.4%). Of these, 215 (87.0%) and 32 (13.0%) reported standardized and non-standardized oral care, respectively. Subsequently, the data from 215 intensive care units that provided standardized oral care were analyzed in detail.

Comment

“Regarding oral care methods, 96 (38.9%) respondents used only a toothbrush, while 116 (46.9%) used both a toothbrush and a non-brushing method.”

How much is the denominator?

It is confusing.

Response

Thank you for your insightful review of our paper. We set “215” as the denominator; however, the denominator was precisely unclear in our abstract. Therefore, we revised our abstract to clearly reflect the denominator as follows:

Page 2, Line 32 – L 36

A total of 609 hospitals and 717 intensive care units nationwide participated; among these, responses were collected from 247 intensive care units (34.4%). Of these, 215 (87.0%) and 32 (13.0%) reported standardized and non-standardized oral care, respectively. Subsequently, the data from 215 intensive care units that provided standardized oral care were analyzed in detail.

Comment

“differed from those in other countries.”

How different?

Response

Thank you for your comment. We revised our abstract based on your comment to make this point clear.

Page 2, Line 39 – L 43

Our study revealed differences in current oral care practices in Japan compared to other countries, particularly the practice of brushing three times a day at unequal intervals and the common use of brushing for oral care. Oral care practices in intensive care units vary worldwide; therefore, more evidence is needed to consider the cultural influences on oral care methods in each country.

Introduction

Comment

“Providing oral care for critically ill patients is crucial for their comfort”

Please provide references.

Response

Thank you for your comment. We provided additional references as follows:

Page 3, Line 50 – L 52

Providing oral care for critically ill patients is crucial for their comfort, improvement of thirst [1,2], and preventing ventilator-associated pneumonia (VAP). Oral care helps improve oral health and reduce inflammation [3,4].

References

Kalfon P, Mimoz O, Auquier P, Loundou A, Gauzit R, Lepape A, et al. Development and validation of a questionnaire for quantitative assessment of perceived discomforts in critically ill patients. Intensive Care Med. 2010;36:1751-1758. doi: 10.1007/s00134-010-1902-9.

Doi S, Nakanishi N, Kawahara Y, Nakayama S. Impact of oral care on thirst perception and dry mouth assessments in intensive care patients: An observational study. Intensive Crit Care Nurs. 2021;66:103073. doi: 10.1016/j.iccn.2021.103073.

Comment

“However, a standardized protocol for oral care is yet to be established, with each hospital following its own protocol.”

This sentence should be moved to the previous paragraph.

Response

Thank you for your insightful review of our paper. We agree with your suggestion. Thus, we have moved the corresponding text to the previous paragraph in the Introduction section. 

Page 3, Line 52 – L 57

Oral care for intubated patients can reduce the incidence of VAP, ventilator days, and length of intensive care unit (ICU) stay [5]. Nurses in many countries recognize the importance of oral care for such patients [6, 7]. However, a standardized protocol for oral care has yet to be established, with each hospital following its own protocol.

 Several countries have conducted studies on the actual situation of oral care in ICUs in the past decade [8, 9].

Materials and Methods

Data analysis

Comment

“The responses to the questionnaire were analyzed by calculating the frequency and percentage of each 97 response.”

What is the difference between the frequency and percentage?

Response

Thank you for your comment. We have revised our manuscript based on the reviewer's comment as follows: 

Page 5, Line 104 – L 107

The responses to the questionnaire were analyzed using descriptive analysis. Categorical variables were calculated as the percentage of each response. For continuous variables, the median and interquartile range (IQR) were calculated for non-normal distributions, and the mean and standard deviation (SD) for normal distributions.

Results

Comment

Participant characteristics

I think information such as gender, experience and designation of the responding nurses is unnecessary for the results and Table1.

Response

Thank you for your comment. Based on your suggestions, we have removed gender, designation, nursing experience, nursing experience at current hospital, and nursing experience in ICU from the Results section and Table 1. In addition, we have revised the name of Table 1, please see Table 1.

Page 6, Line 128 – L 131

Characteristics of participating ICUs are listed in Table 1. Regarding participating hospitals, most respondent hospitals had approximately 401–600 beds. In addition, the respondent hospitals had a median of 8 (IQR 6–12) and 7.5 (IQR 6–10) ICU beds for the standardized and non-standardized groups, respectively.

Page 6, Line 134

Table 1. Characteristics of participating ICUs

Comment

Table 1. Participant characteristics

“Intensivist is the attending physician”, “Intensivists are not attending physicians; however, all patients admitted to the ICU are seen by an intensivist”, “Intensivists are involved in patient care only when the attending physician requests to see them”

The explanation is a bit redundant; why not state closed ICU (J Intensive Care. 2018 Sep 3:6:57.), semi closed ICU, open ICU, etc., and put the explanation below the table?

Response

 Thank you for your insightful comment. We have reviewed the reference reviewer provided and we agree with your suggestions. Thus, we revised Table 1 based on your suggestions, please see Table 1. 

Page 7, Line 136 – L139

a　Intensivist is the attending physician.

b　Intensivists are not attending physicians; however, all patients admitted to the ICU are visited by an intensivist.

c　Intensivists are involved in patient care only when the attending physician requests to see them.

Comment

Table 2. Response to oral condition assessment

“Frequency”

The number of the hospital (%)?

Response

 Thank you for pointing this out. The column heading “Frequency” was unclear, and we have revised it to “Value (n = 215 ICUs)”. Please see Table 2.

Comment

Table 3. Response to toothbrushing and other methods of oral care

Same as above.

Response

Thank you for pointing this out. The column heading “Frequency” was unclear, and we have revised it to “Value (n = 212 ICUs)”. Please see Table 3.

Discussion

Comment

“Furthermore, the differences and similarities between Japanese ICUs and those in other countries were highlighted.”

The differences between Japanese ICUs and ICUs in other countries was described, but what are the similarities?

Response

We appreciate your helpful comment. There are similarities in the insufficient repeated use of assessment tools after ICU admission between Japan and other countries compared to the international consensus. We have added to the discussion section to make this point clear.

Page 11, Line 199 - 204 

A study revealed that 35% of nurses in the United Kingdom used oral assessment tools during ICU admission [14], and another study reported that most nurses in the United States did not use oral assessment tools [8]. The current state of oral care in Japan, as found in our study, shows the difference in the rate of use of oral assessment tools compared to that reported in the United Kingdom and the United States. However, there may be insufficient repeated use of assessment tools compared to international consensus, the same as the United Kingdom and the United States.

Comment

“However, our findings may be insufficient according to international consensus.”

It is difficult to understand the meaning of this sentence.

Do authors mean “However, the current state of oral care in Japan, as found in our study, may be insufficient according to the international consensus.”

Response

 Thank you for your helpful comment. We have revised our manuscript in response to the reviewer’s suggestion as follows:

Page 11, Line 199 - 204

A study revealed that 35% of nurses in the United Kingdom used oral assessment tools during ICU admission [14], and another study reported that most nurses in the United States did not use oral assessment tools [8]. The current state of oral care in Japan, as found in our study, shows the difference in the rate of use of oral assessment tools compared to that reported in the United Kingdom and the United States. However, there may be insufficient repeated use of assessment tools compared to international consensus, the same as the United Kingdom and the United States.

---

## [Decision Letter · Decision Letter 2]

16 Feb 2024

PONE-D-23-23882R2Nation-Wide Survey of Oral Care Practice in Japanese Intensive Care Unit : a Descriptive StudyPLOS ONE

Dear Dr. Unoki,

Thank you for submitting your manuscript to PLOS ONE. After careful consideration, we feel that it has merit but does not fully meet PLOS ONE’s publication criteria as it currently stands. Therefore, we invite you to submit a revised version of the manuscript that addresses the points raised during the review process.

Dear Author,

**Thank you for resubmitting the manuscript.**

**Reviewer raise some concerns, and I please revise accrding the reviewers comments.**

**Especially,**

**・"The first paragraph of a discussion should generally describe what was found by the study. "**

**If you cannot fully responce the comment by reviewer, you can add the limitation of your study.**

We look forward to receiving your revised manuscript.

Kind regards,

Yujiro Matsuishi

Academic Editor

PLOS ONE

Additional Editor Comments:

Dear Author,

Thank you for resubmitting the manuscript.

Reviewer raise some concerns, and I please revise accrding the reviewers comments.

Especially,

・"The first paragraph of a discussion should generally describe what was found by the study. "

If you cannot fully responce the comment by reviewer, you can add the limitation of your study.

Reviewers' comments:

Reviewer's Responses to Questions

**Comments to the Author**

1. If the authors have adequately addressed your comments raised in a previous round of review and you feel that this manuscript is now acceptable for publication, you may indicate that here to bypass the “Comments to the Author” section, enter your conflict of interest statement in the “Confidential to Editor” section, and submit your "Accept" recommendation.

Reviewer #4: All comments have been addressed

2. Is the manuscript technically sound, and do the data support the conclusions?

Reviewer #4: Yes

3. Has the statistical analysis been performed appropriately and rigorously? 

Reviewer #4: Yes

4. Have the authors made all data underlying the findings in their manuscript fully available?

Reviewer #4: Yes

5. Is the manuscript presented in an intelligible fashion and written in standard English?

Reviewer #4: Yes

6. Review Comments to the Author

Reviewer #4: Thank you for allowing me to review "Nation-Wide Survey of Oral Care Practice in Japanese Intensive Care Unit : a Descriptive Study" by Kuribara, Unoki et al.

The authors addressed the reviewer’s comments.

As stated in the previous review, it is difficult to see whether this study is appropriate to be published as a full paper. I leave it to the editorial board to decide, but a letter might be more appropriate.

My comments are as follows.

Abstract

“Our study revealed differences in current oral care practices in Japan compared to other countries, particularly the practice of brushing three times a day at unequal intervals and the common use of brushing for oral care.”

This study only reports the results of a questionnaire on the current situation in Japan and does not identify any differences between Japan and other countries. This statement is therefore inappropriate and needs to be corrected.

Also, my question, ' How different?' is not answered.

Discussion

“Furthermore, the differences and similarities between Japanese ICUs and those in other countries were highlighted.”

The authors mentioned similarity, but I still don't think it is an appropriate sentence to include in the first paragraph of a discussion. The first paragraph of a discussion should generally describe what was found by the study. For the readers, the difference and similarity are not highlighted at this point. It should be explained in detail in the paragraphs to come.

7. PLOS authors have the option to publish the peer review history of their article (what does this mean?). If published, this will include your full peer review and any attached files.

Reviewer #4: No

---

## [Author Response · Author response to Decision Letter 2]

12 Mar 2024

Response to Reviewers

Dear Editor and Reviewers:

Thank you very much for reviewing our manuscript and offering valuable advice. We have addressed your comments with point-by-point responses and revised the manuscript accordingly.

Comments from Academic Editor 

Thank you for resubmitting the manuscript.

Reviewer raise some concerns, and I please revise accrding the reviewers comments.

Especially,

・"The first paragraph of a discussion should generally describe what was found by the study. "

If you cannot fully responce the comment by reviewer, you can add the limitation of your study.

Response

 Thank you for the valuable advice and recommendations. We have revised our manuscript according to each comment. Our point-by-point responses are given below.

Comments from Reviewer #4:

Thank you for allowing me to review "Nation-Wide Survey of Oral Care Practice in Japanese Intensive Care Unit : a Descriptive Study" by Kuribara, Unoki et al.

The authors addressed the reviewer’s comments.

As stated in the previous review, it is difficult to see whether this study is appropriate to be published as a full paper. I leave it to the editorial board to decide, but a letter might be more appropriate.

My comments are as follows.

Response

 We appreciate your helpful comments regarding our manuscript. We have revised our manuscript according to your comments and provided point-by-point responses below.

Comment

Abstract

“Our study revealed differences in current oral care practices in Japan compared to other countries, particularly the practice of brushing three times a day at unequal intervals and the common use of brushing for oral care.”

This study only reports the results of a questionnaire on the current situation in Japan and does not identify any differences between Japan and other countries. This statement is therefore inappropriate and needs to be corrected.

Also, my question, ' How different?' is not answered.

Response

Thank you for your valuable insights. We sincerely apologize for our inadequate responses and revisions in the previous round. Although this study has clarified the differences between ICUs in Japan and those in other countries through its discussion, as you point out, this study only reports the results of a questionnaire on the current situation in Japan and does not identify any differences between Japan and other countries through its findings. Therefore, we have modified the aim and conclusions in the Abstract and main text. We have also made some changes, visible in red, throughout the manuscript; these revisions emphasize that the differences being discussed are specific to ICUs in Japan and those in other countries.

Page 2, Lines 27–29

Thus, this study aimed to determine the overall oral care practices, including those for intubated patients, in Japanese intensive care units. We also discuss the differences in oral care methods between Japanese ICUs and ICUs in other countries.

Page 2, Lines 41–44

The findings of our study reveal current oral care practices in ICUs in Japan. In particular, most ICUs provide oral care three times a day at unequal intervals, and almost all use toothbrushes as a common tool for oral care. The results suggest that some oral care practices in Japanese ICUs differ from those in ICUs in other countries.

Page 3, Lines 67–70 

Therefore, this study aimed to determine the overall oral care practices, including those for intubated patients, in ICUs throughout Japan. We also discuss the differences in oral care methods between Japanese ICUs and ICUs in other countries.

Page 13, Lines 253–256

 In summary, most ICUs in Japan provide oral care three times a day at unequal intervals, and almost all use toothbrushes as a common tool for oral care. The results suggest that some oral care practices in Japanese ICUs differ from those in ICUs in other countries. Future studies should focus on the efficacy of different oral care methods to provide more evidence.

Comment

Discussion

“Furthermore, the differences and similarities between Japanese ICUs and those in other countries were highlighted.”

The authors mentioned similarity, but I still don't think it is an appropriate sentence to include in the first paragraph of a discussion. The first paragraph of a discussion should generally describe what was found by the study. For the readers, the difference and similarity are not highlighted at this point. It should be explained in detail in the paragraphs to come.

Response

 Thank you for your valuable comments. We agree with your suggestion and have modified the first paragraph of the Discussion to show the main findings of the study.

Page 11, Line 191–195

The findings of this study revealed current oral care practices in ICUs in Japan. In particular, most ICUs provided oral care three times a day at unequal intervals, mainly in the morning, daytime, and evening, and almost all used toothbrushes as a common tool for oral care, with water being the predominant choice for rinsing the mouth after brushing. Approximately half of ICUs used oral assessment tools or provided moisturizing care in addition to toothbrushes.

---

## [Editor Report · Decision Letter 3]

14 Mar 2024

Nation-wide survey of oral care practice in Japanese intensive care units: A descriptive study

Short title: Oral care practice in Japanese ICUs

PONE-D-23-23882R3

Dear Dr. Unoki,

We’re pleased to inform you that your manuscript has been judged scientifically suitable for publication and will be formally accepted for publication once it meets all outstanding technical requirements.

Kind regards,

Yujiro Matsuishi

Academic Editor

PLOS ONE

Additional Editor Comments (optional):

Dear authors.

Thank you for your resubmission.

The authors have responded satisfactorily and we have decided to accept the manuscript at this stage.

We thank you for your submission to PLOS ONE.

Yours sincerely

---

## [Editor Report · Acceptance letter]

21 Mar 2024

PONE-D-23-23882R3 

PLOS ONE

Dear Dr. Unoki, 

I'm pleased to inform you that your manuscript has been deemed suitable for publication in PLOS ONE. Congratulations! Your manuscript is now being handed over to our production team.

Kind regards, 

on behalf of

Dr. Yujiro Matsuishi 

Academic Editor

PLOS ONE